# Methods of the Dehydration Process and Its Effect on the Physicochemical Properties of Stingless Bee Honey: A Review

**DOI:** 10.3390/molecules27217243

**Published:** 2022-10-25

**Authors:** Liyana Nabihah Ikhsan, Kok-Yong Chin, Fairus Ahmad

**Affiliations:** 1Department of Anatomy, Faculty of Medicine, Universiti Kebangsaan Malaysia Medical Centre, Jalan Yaacob Latif, Bandar Tun Razak, Kuala Lumpur 56000, Malaysia; 2Department of Pharmacology, Faculty of Medicine, Universiti Kebangsaan Malaysia Medical Centre, Jalan Yaacob Latif, Bandar Tun Razak, Kuala Lumpur 56000, Malaysia

**Keywords:** stingless bee honey, dehydrated honey, chemical composition, dehydration method

## Abstract

Stingless bee honey (SLBH) has a high moisture content, making it more prone to fermentation and leading to honey spoilage. Dehydration of SLBH after harvest is needed to reduce the moisture content. This review compiles the available data on the dehydration methods for SLBH and their effect on its physicochemical properties. This review discovered the dehydration process of vacuum drying at 60 °C and 5% moisture setting, freeze-drying at −54 °C and 5% moisture setting for 24 h, and using a food dehydrator at 55 °C for 18 h could extract >80% water content in SLBH. As a result, these methods could decrease moisture content to <17% and water activity to <0.6. These will prevent the fermentation process and microorganism growth. The hydroxymethylfurfural (HMF) contents remain within the permissible standard of <40 mg/kg. The total phenolic content increased after dehydration by these methods. Therefore, dehydration of SLBH is recommended to increase its benefits.

## 1. Introduction

Stingless bee honey (SLBH) is an emerging functional food due to its many health benefits. SLBH is rich in flavonoid and phenolic content, which contributes to its high antioxidant activity [1]. The common phenolic compounds in SLBH are the same as *Apis mellifera*, such as salicylic acid, p-coumaric acid, caffeic acid, chlorogenic acid, ferulic acid and quercetin [2]. Despite its benefits, SLBH has generally higher moisture content than *Apis* spp. [3]. A previous study showed that the moisture content in SLBH is the highest with 33.24%, compared to *Apis* spp. which is between 21.96–27.41% [4]. SLBH of *Melipona* spp. has a moisture content above 24.8% compared to *Apis* spp. with 18.6% [5]. A comprehensive review reported that SLBH contains more moisture (21.52–31%) than Tualang and Gelam honey (17.53–26.51%) [6]. Furthermore, SLBH has a high water activity of 0.76 compared to a range between 0.60–0.67 in *Apis* spp. [4].

The high moisture content in SLBH makes it more susceptible to alcoholic fermentation contributing to honey acidity [7]. The rapid fermentation by microorganism growth in SLBH leads to honey spoilage [8]. Apart from high water content, SLBH has higher free acidity, electrical conductivity and lower diastase activity compared to *Apis* spp. [3]. Hence, it is difficult for SLBH to follow the honey standard. Several studies have proposed a different standard for SLBH given the difficulty for SLBH to follow the International Honey Commission (IHC) standard [9].

Owing to the high water content, it is a challenge to maintain the quality of SLBH. Therefore, dehydration of SLBH upon collection is suggested to lower the moisture content. Microbial stability can be achieved through dehydration, thus, prolonging the shelf life of honey [8]. Additionally, lowering the moisture content will help the SLBH adhere to the standard. However, previous studies have shown that the dehydration process can reduce the phenolic content [10]. In addition, thermal treatment can increase hydroxymethylfurfural (HMF) content in honey [11]. The phenolic content is an essential source of antioxidants in honey [1]. Meanwhile, HMF is a potential carcinogenic and genotoxic agent [12]. Therefore, a suitable dehydration method is needed to obtain the maximum benefit from SLBH by reducing its moisture content without compromising its phenolic content and ensuring a safe level of HMF.

Globally, almost 500 species of SLBH are distributed in South America, Africa, Australia and Southeast Asia [13]. Despite the numerous species, the most commonly domesticated stingless bee honeys by beekeepers worldwide are from *Melipona* and *Trigona* genera [14]. To the best of our knowledge, there are limited publications on the dehydration of SLBH that provide information on the changes in its physicochemical properties. This review aims to provide an overview of the available information on the physicochemical properties of SLBH before and after the dehydration process. This will help to determine the most optimal setting and method of dehydration for SLBH without compromising its benefit. The physicochemical information retrieved includes moisture content, water activity, pH, free acidity, hydroxymethylfurfural (HMF), ash, electrical conductivity, diastase, sugar content, total soluble solids, total phenolic content and total flavonoid content. This review also compiles the effect of dehydration on individual phenolic compounds. The dehydration of SLBH that was conducted to reduce the moisture content includes thermal treatment, thermosonication, vacuum drying, vacuum evaporation, freeze-drying, microwave heating, dehumidification, food dehydrator, Malaysian Agricultural Research and Development Institute (MARDI) dehydrator and passive diffusion [15,16,17,18,19,20,21,22,23,24]. We hope this review could give better insight and aid the readers in deciding the most effective dehydration method to maximize the benefits of SLBH.

## 2. Physicochemical Properties of Dehydrated Stingless Bee Honey

### 2.1. Moisture Content

Moisture content is the amount or percentage of water present in the honey [25]. Water in honey is the key factor for honey quality as it determines the ability of honey to resist spoilage by microorganism fermentation [26]. Previous studies presented in Table 1 showed that the percentage of moisture content of raw SLBH was between 23.9 and 40%. However, another study showed that the moisture content of raw SLBH was between 13.26 and 45.8% [9]. The wide range in the percentage of moisture content was due to environmental factors such as seasonal weather and humidity [8]. Harvest and storage conditions also influenced the moisture content in SLBH [9].

Several studies summarized in Table 1 showed that reduction in the water content of the SLBH after harvesting could be achieved either by increasing the temperature through various dehydration methods or via passive diffusion. The temperature used in the dehydration process was between 30 and 95 °C, while the temperature for the passive diffusion method was between 25 and 35 °C. As a result, the moisture content of SLBH was reduced between 29.6 and 5% after the dehydration process using these various methods. Moisture content below 17% could prevent the fermentation process by the microorganisms [27].

As a conclusion, according to the data presented in Table 1, the dehydration process using thermal treatment will only cause less than a 10% reduction in water content. Meanwhile, another study showed that the thermosonication method of the dehydration process caused a 16.6% reduction in water content compared to 6.9% using the thermal method [22]. These findings suggest that thermosonication is the better method for the dehydration process for SLBH compared to thermal treatment. However, both methods could not reduce the moisture content below 17% (25.9% for thermosonication and 28.8% for the thermal treatment method).

A study showed that the moisture content of the SLBH was reduced from 31.9 to 11 and 5% after the dehydration process using vacuum and freeze-drying methods [21]. As presented in Table 1, the vacuum drying and freeze-drying at 5% moisture setting could achieve an 84.3% reduction in water content. Meanwhile, a 65.5% reduction in water content of the SLBH was observed after dehydration using vacuum drying and evaporation at 11% moisture setting. These findings suggest that both vacuum treatment and freeze-drying are the best methods in reducing the moisture content of SLBH. In addition, both methods could achieve a safe level of moisture content below 17%.

A study by Yegge et al. [18] showed that the dehydration process using microwave heating and dehumidification methods could reduce water content by up to 52% and 45%, respectively, as presented in Table 1. In the study, both methods could reduce almost half of the water content in raw SLBH. The microwave heating method used a power level of energy (PL) of 20, 60 and 100. However, only the microwave heating method at 60 PL for 60 s could reduce the moisture content below 17% (from 31.47 to 15.04%). Meanwhile, the dehumidification process was performed for 1 to 2 days. Therefore, microwave heating at 60 PL for 1 min was the best method for achieving the recommended moisture content level below 17%. In addition, this method was more practical because it takes less time to prepare the dehydrated SLBH.

From the data provided in Table 1, a previous study has also shown that the dehydration process of SLBH using a food dehydrator could reduce the water content of SLBH up to 80–100% [17]. The food dehydrator could achieve 80% water reduction at 40 °C for 36 h or at 55 °C and 70 °C for 18 h. Complete water reduction was achieved at 55 °C and 70 °C by prolonging the duration of the dehydration process to 36 h [17]. Another study showed that a dehydrator developed by the Malaysian Agricultural Research and Development Institute (MARDI) could reduce 35% of the water content [24]. However, the MARDI dehydrator set at 30 °C for 8 h was unable to reduce the moisture content below 17% [24]. Meanwhile, the conventional food dehydrator set between 40 and 70 °C for the duration of 18 to 36 h could achieve recommended moisture content level below 17% [17]. These findings suggest that a higher temperature would result in a higher reduction in moisture content.

Several studies summarized in Table 1 showed that the dehydration process of the SLBH can be performed via passive diffusion by storage in a clay pot. A study by Ghazali et al. [19] showed that the reduction in the moisture content was significant in the clay pot compared to the glass container. In addition, the storage in a clay pot with a larger surface area resulted in a 10.9% reduction in water content compared to a smaller clay pot with only 7.21%. This finding suggests that the larger the surface area of the container, the more effective the passive diffusion process will occur. On the other hand, the storage of SLBH at 35 °C for three days could reduce up to 24.2% of water [20]. Meanwhile, the storage of SLBH at room temperature (25 °C) for 21 days could reduce water content by up to 29.8% content [20]. These findings suggest a higher temperature would expedite the passive diffusion process. However, the dehydration process via passive diffusion requires a long duration to reduce the moisture content of the SLBH. Furthermore, the moisture content after storage in the clay pot was between 18.13 and 25.13%, which was still above the recommended moisture content level at 17%.

Various dehydration methods of SLBH can reduce moisture content depending on the temperature and duration of the dehydration process. We concluded that the higher the temperature setting, the more reduction in water content. For that, we suggest the dehydration method of SLBH at a high temperature setting to achieve at least less than 17% moisture content to retard the fermentation process. The methods that yield low moisture contents are vacuum treatment, freeze-drying, food dehydrator and microwave heating at 60 PL. In conclusion, we observed that the food dehydrator is the best method because it could remove up to 80 to 100% water content, resulting in moisture content of less than 17%. However, it takes up to 18 to 36 h in duration. Therefore, microwave heating at 60 PL is the method of choice due to the short duration of 60 s with the moisture content of less than 17%. Although the vacuum treatment could reduce the moisture content to 5 and 11%, the duration of the dehydration process was not mentioned by the authors.

### 2.2. Water Activity

Water activity is a measurement of free unbound water that can be utilized by microorganisms for growth [28]. Water activity gives a better prediction of the likelihood of the fermentation process occurring compared to moisture content [26]. Therefore, water activity is used as an indicator of food stability, which is important for the determination of honey spoilage due to microbial growth [4]. Water activity (aw) is expressed in decimals and calculated from equilibrium relative humidity (ERH) divided by 100 (aw = ERH (%)/100) [29]. ERH is the equilibrium of humidity of the food product with its environment.

Microorganisms will not grow below a particular water activity level, which is 0.90 for bacteria and 0.70 for molds. A water activity of less than 0.6 will halt all types of microbial growth [28]. Hence, it is crucial to maintain the water activity of SLBH below 0.6. Water activity is strongly correlated with moisture content [26]. Therefore, the dehydration process of SLBH is needed to reduce moisture content and water activity in SLBH. Subsequently, the dehydration process will help to prevent the likelihood of fermentation due to the inability of the microorganism to grow in the SLBH. A previous study reported that SLBH has the highest water activity of 0.76 compared to *Apis* spp. and commercialized honey with water activity ranges between 0.54–0.67 [4]. Several studies compiled in Table 2 showed that the water activity of raw SLBH was between 0.79 and 0.807. After the dehydration process, the water activity was reduced between 0.28 and 0.785 as presented in Table 2.

According to the data summarized in Table 2, thermosonication causes a 7.9% reduction in water activity compared to 3.5% for the thermal method [22]. This suggests that thermosonication is the better method in reducing water activity compared to thermal treatment. However, the water activity was 0.743 and 0.767 for thermosonication and thermal treatment, respectively, which was still above 0.6.

A study by Chen et al. [21] showed that dehydration processes using vacuum and freeze-drying methods at a 5% moisture setting were able to reduce the water activity level from 0.79 to less than 0.3, as presented in Table 2. Meanwhile, the water activity level of the vacuum drying and evaporation at an 11% moisture setting could reduce the water activity level to less than 0.5. These findings suggest that both vacuum treatment and freeze-drying methods could reduce the water activity level of SLBH to less than 0.6. It is observed that vacuum drying and freeze-drying at a 5% moisture setting was the best dehydration process for reducing the water activity level. However, the freeze-drying method at a 5% moisture setting needs 24 h to achieve a 0.3 water activity level. Meanwhile, the duration of vacuum treatment was not mentioned by the author.

A study showed that dehydration of SLBH using a food dehydrator could reduce water activity levels from 0.788 to less than 0.6 [17]. In the study, the water activity of less than 0.6 was achieved with 40 °C for 36 h, 55 °C for 18 h, and 70 °C for 12 h, as summarized in Table 2. The study findings showed that the dehydration process using a food dehydrator at a higher temperature will take less time to reduce the water activity level to less than 0.6. 

From the data provided in Table 2, the dehydration of SLBH via passive diffusion could reduce water activity levels from a range between 0.79–0.8 to 0.63–0.785 [19]. The study showed that SLBH stored in a clay pot could reduce water activity up to 21% compared to storage in a glass container, which was only up to 2.25%. The surface area of the clay pot also plays an important role in the reduction in water activity. The study showed up to 21% reduction in the water activity level in a clay pot with a larger surface area compared to 15.1% for a clay pot with a smaller surface area [19]. Another study has also shown more reduction in water activity will be achieved in a clay pot at 35 °C compared to 25 °C [20]. In the study, the water activity level of the SLBH in a clay pot at 35 °C was 0.7 after three days. Meanwhile, the water activity level of the SLBH in a clay pot at 25 °C was 0.7 after seven days. Therefore, the duration of the dehydration process is shorter as the temperature setting in the clay pot storage increases. However, this passive diffusion method of the dehydration process was unable to reduce the water activity level below 0.6.

All the dehydration methods could reduce water activity levels. In addition, a higher temperature setting of the dehydration process progressively reduces the water activity level of the SLBH. The methods of dehydration that produced a water activity level below 0.6 were vacuum method, freeze-drying, and food dehydrator. The food dehydrator at 70 °C was the best method of dehydration to achieve a water activity level below 0.6 within a shorter duration. Meanwhile, the vacuum method could reduce water activity levels below 0.5, but the duration was not mentioned by the authors.

### 2.3. Hydroxymethylfurfural

Hydroxymethylfurfural (HMF) is a chemical compound from the furan group that indicates the freshness, overheating and ageing of honey [7]. HMF and diastase activity are indicators of overheating. However, HMF is a more reliable parameter for overheating compared to diastase activity [30]. The HMF content can provide information regarding total heat exposure to honey [30]. Apart from overheating, prolonged honey storage also promotes the formation of HMF by degradation of the honey sugar into HMF [31]. A previous study showed that raw SLBH has a lower HMF content than *Apis mellifera* honey because raw SLBH has higher acidity and water activity that can slow down the Maillard reaction [32].

Codex Alimentarius Standards (2001) has set that the HMF level should not exceed 40 mg/kg for honey, except for that from the tropical region, which should not exceed 80 mg/kg. A high concentration of HMF is potentially carcinogenic and genotoxic [12]. A previous study showed that the heat from thermal treatment can increase the HMF content in honey [11]. Therefore, the dehydration process needs to be controlled to ensure strict adherence to the maximum permitted amount of HMF. Several studies summarized in Table 3 showed that the HMF was not detected in raw SLBH, except for in a study by Syariffuddeen et al. [24] that reported a HMF level of 2.27 mg/kg. After the dehydration process, the HMF content either remained unchanged or increased between 2.39 and 238.18 mg/kg, as presented in Table 3.

According to the data provided in Table 3, the HMF content remained below the detection level after thermal treatment. However, as the duration of thermal treatment was prolonged at 75 °C for 24 h, the HMF content increased to 238.18 mg/kg, which exceeded the standard set by Codex [32]. Similarly, a study by Chong et al. [22] showed that the HMF level remained undetected at a low temperature and short duration in the thermal treatment and thermosonication method. However, the HMF content increased as the temperature and duration of dehydration increased to 67.5–90 °C for 100–120 min. HMF level was higher after the thermosonication process (62.46 mg/kg) compared to in the thermal treatment (42.40 mg/kg). These findings suggest that the thermal treatment is the better dehydration method for SLBH compared to thermosonication. However, prolonged duration and higher temperature setting in both methods can increase HMF level above the permitted level of 40 mg/kg.

A previous study showed that HMF content increased after the dehydration process using vacuum drying and freeze-drying methods from zero to up to 12.18 mg/kg, and 9.29 mg/kg, respectively [21]. The HMF content increased as the temperature of vacuum drying increased, as shown in Table 3. Both dehydration methods were able to increase the HMF content even at low temperatures, as low as 40 and −54 °C. However, the increase in HMF content was far below the permitted level of 40 mg/kg.

A study by Yap and colleagues [17] showed that HMF content remained undetected after the dehydration process using a food dehydrator at low temperature as presented in Table 3. However, as the temperature and duration increased to 55 and 70 °C for 36 h, the HMF content increased to 5.81 and 83.19 mg/kg, respectively. These findings suggest that the HMF content increases along with the increase in temperature and duration of the dehydration process. On the other hand, the dehydration process using the MARDI dehydrator at 30 °C for 8 h showed a slight increase in HMF from 2.27 to 2.39 mg/kg [24]. Therefore, the settings in which HMF content remained below 40 mg/kg were a food dehydrator at 40 and 50 °C for 18 to 36 h and the MARDI dehydrator. The HMF content exceeded the permissible limit using a food dehydrator above 70 °C.

In conclusion, the HMF content increased as the temperature and duration of the dehydration process increased. These were observed in thermal treatment, thermosonication, vacuum drying and dehydration using food dehydrator methods. The dehydration method that could maintain HMF content below the permitted level of 40 mg/kg were the thermal treatment for a short duration, thermosonication at 45–67.5 °C for 30–75 min duration, vacuum drying, freeze-drying, food dehydrator at 40 and 55 °C for 18–36 h, and MARDI dehydrator. The best dehydration method was food dehydrator at 40 °C for 36 h duration setting, because the HMF content remained undetectable, although heated for many hours.

### 2.4. pH

Honey is naturally acidic due to the presence of organic acids [25]. SLBH has a mixed sweet and sour taste [33]. Previous studies reported that SLBH has the lowest pH of 3.04 compared to Tualang and Acacia honey, with pH of 3.63 and 3.61, respectively [34]. Honey becomes more acidic as a result of the fermentation process [25]. Therefore, the rapid susceptibility of SLBH to alcoholic fermentation will further reduce the pH level. Hence, dehydration of honey is required for water removal, which subsequently, can prevent honey fermentation. Various studies compiled in Table 4 showed that the pH of raw SLBH ranges from 3.11 to 5.18. These values were consistent with a previous study that showed that the pH level of raw SLBH ranges between 3.15 and 6.64 [9]. After the dehydration process, the pH value ranges between 3.11 and 5.14, as shown in Table 4.

Several studies presented in Table 4 showed that the dehydration process by thermal treatment could maintain or decrease the pH level. Meanwhile, another study showed that thermal treatment at 75 °C and 90 °C could increase the pH value by 2.1% and 3.9%, respectively [23]. These findings suggest that the dehydration process of SLBH at a high temperature could improve the pH level, making SLBH less acidic. However, the duration was not mentioned by the authors.

The dehydration method by vacuum drying at 5% moisture content could increase pH by 7.9% compared to 4.7% in vacuum drying at 11% moisture content, as shown in Table 4 [21]. Meanwhile, no significant changes were observed in pH level after the dehydration process by vacuum evaporation and freeze-drying method [21]. Therefore, vacuum drying was the best method because it could improve the pH value.

A study by Yegge et al. [18] showed that the dehydration process using microwave heating decreased the pH value from 3.58 to a range between 3.45 and 3.51. This could be attributed to the microwave heating pressure that degrades the honey into organic acids, which gives rise to acidity [18]. The same study showed that dehydration of SLBH by the dehumidification process initially decreased pH value on day 1; subsequently, the pH value increased on day two of treatment, as presented in Table 4. However, the improvement in pH was not significant and the dehumidification process had a long duration, up to days. Hence, we observed that both microwave heating and dehumidification methods did not improve the pH value by much.

According to the data provided in Table 4, dehydration of SLBH via passive diffusion by a clay pot storage for 10 days and storage of SLBH in a glass container did not alter the pH value [19]. Both samples were stored at 25 °C. This indicates that no fermentation process occurred. In contrast, a study by Baroyi et al. [20] showed that storage of SLBH in a clay pot at 25 °C for seven days and 35 °C for three days increased the pH level by 5.2 and 5.5%, respectively. These findings suggest that storage of SLBH in a clay pot at a higher temperature could improve the pH level better than storage at a lower temperature. However, the pH value started to deteriorate when stored for longer than 14 days at 25 °C [20]. This scenario might be due to a successful dehydration process during the early days of storage that improved the pH value. However, as the duration of storage was prolonged for more than two weeks, the fermentation process started to occur, which reduced the pH level. These suggest that the passive diffusion method is able to improve the pH level during the early days of storage and cannot further improve the pH level as soon as the fermentation process has occurred. We observed that the dehydration via passive diffusion method by a clay pot storage at higher temperature of 35 °C was the better method compared to other passive diffusion dehydration settings. In this setting, it could achieve the highest increase in pH, up to 5.5% within three days.

In conclusion, the dehydration process of SLBH can raise, maintain or reduce the pH level. Despite successful water removal through the dehydration process, it cannot ensure that the pH of SLBH will improve. Methods of dehydration that could increase the pH level are thermal treatment at higher temperature, vacuum drying, and passive diffusion by storage in a clay pot at 25 °C for seven days and at 35 °C for three days. The best dehydration method is vacuum drying at a 5% moisture setting because it resulted in the highest improvement in pH value, which was 7.9%.

### 2.5. Free Acidity

Free acidity is an indicator of organic acids in honey [9]. The elevation of the free acidity level indicates that the fermentation process from sugar to organic acids has occurred [35]. Free acidity is expressed in milliequivalents of acid per one kilogram honey (meq/kg) unit. A previous study reported that SLBH has a high free acidity level that is 2.7 times higher than *Apis* spp., Manuka and commercialized honey [4].

Earlier studies have shown that the free acidity of raw SLBH was between 23.2 and 172 meq/kg, as summarized in Table 5. Meanwhile, another study showed that the free acidity of raw SLBH ranges between 5.9 and 592 meq/kg [9]. The wide variations in free acidity levels are attributed to the differences in organic acids, floral and geographical origin [36]. Following the dehydration process, the free acidity ranges from 21.3 to 181 meq/kg, as shown in Table 5.

According to the data compiled in Table 5, the thermal treatment could reduce the free acidity level. However, thermal treatment at low temperature for a long duration or at high temperature for a short duration did not affect the free acidity level in SLBH. These findings suggest that organic acids can be preserved with thermal treatment. Otherwise, the thermal treatment may reduce the organic acid content in SLBH.

A study by Chen et al. [21] showed that the dehydration process by vacuum methods decreased the free acidity level by up to a 26.2% reduction, as presented in Table 5. Meanwhile, the freeze-drying method did not affect the free acidity level [21]. The temperature setting in the vacuum methods was at 40 to 60 °C while freeze-drying was at −54 °C. Free acidity reduced in the vacuum methods because the heat from the vacuum promoted the loss of volatile organic acids by evaporation or decomposition [21]. Therefore, freeze-drying was the better method compared to vacuum methods given its ability to preserve the content of organic acid after the dehydration process. This was because the freeze-drying method was conducted at a very low temperature. Hence, the SLBH was not exposed to heat.

Previous studies summarized in Table 5 showed that the dehydration method by passive diffusion via storage in a clay pot could maintain the free acidity level when stored at 25 °C for 10 days and at 35 °C for three days. These findings suggest that the fermentation process did not occur when SLBH was stored at both storage conditions. In contrast, a study by Baroyi et al. [20] showed that storage of SLBH in a clay pot at 25 °C increased the free acidity up to 16.5% during the first seven days of storage, while it was decreased after 14–21 days of storage. These findings suggest that the fermentation process may occur as early as the first week of storage, as evidenced by an increase in the free acidity level. After 2–3 weeks of storage, the organic acid content in SLBH started to decline, evidenced by a decrease in the free acidity level. Overall, we observed that storage of SLBH in a clay pot resulted in various outcomes with regards to the free acidity. This might be due to the nature of the dehydration method via passive diffusion method, which took days. The prolonged storage duration increases the possibility of the fermentation process to occur.

Some dehydration methods could preserve the organic acid in SLBH by maintaining the free acidity level. However, it depends on the temperature used during the dehydration process, in which less exposure to the heat treatment preserves the free acidity. The dehydration methods that can maintain the free acidity level are thermal treatment at low temperature for a long duration or at high temperature for a short duration, freeze-drying, and passive diffusion at 35 °C for a three day storage period. We observed that the best dehydration method was the freeze-drying method at a 5% moisture setting because it was performed at a low temperature. Hence, the lack of heat exposure limits the possibility of organic acid reduction in SLBH.

### 2.6. Ash

According to the International Honey Commission (IHC, 2009), the ash content of honey is obtained by a defined procedure and expressed as the percentage by weight. The honey is ashed at a temperature no higher than 600 °C and the residue is weighed. Ash content in honey is important as it indicates the mineral content [9]. IHC (2009) established a guideline for ash content of not more than 0.5 g/100 g, whereas the Malaysian Standard (2017) sets a higher value, which is not more than 1.0 g/100 g.

Previous studies presented in Table 6 showed that the percentage of the ash content of the raw SLBH was between 0.055 and 0.11 g/100 g. However, another study by Nordin et al. [9] showed that the ash content of raw SLBH was between 0.01 and 3.1 g/100 g. The wide variation of ash content is influenced by the composition of plant nectar [9]. After the dehydration process via passive diffusion by storage in a clay pot, the ash content was able to be maintained or increased between 0.051 and 0.18 g/100 g, as shown in Table 6. Reduced moisture following dehydration caused more concentrated mineral content, which led to an increase in ash level [19].

A previous study by Ghazali et al. [19] compared the alteration of ash content when SLBH was stored in a glass bottle and clay pots of different sizes at room temperature. The ash content was maintained throughout the storage in a glass bottle. This finding suggests that no water loss occurred in glass bottle storage, hence, no significant effect on ash content. Meanwhile, a clay pot with large surface area was able to increase the ash content by 65.5% as early as day seven of storage, compared to 60% by day 10 of storage in a smaller clay pot, as shown in Table 6. These findings suggest that SLBH stored in a larger clay pot took less time to lose more water than a smaller clay pot, resulting in higher ash concentrations. Another study by Baroyi et al. [20] showed no significant changes when SLBH was stored at different storage temperature. These findings suggest that mineral content in SLBH is similar to raw SLBH regardless of the difference in storage temperature.

In conclusion, the dehydration method via passive diffusion by a clay pot could promote water removal, subsequently, preserving or increasing the ash content. We observed that a clay pot with large surface area is the best dehydration method via passive diffusion with regards to ash content. This is because it takes less time to promote more water removal and, subsequently, increase the ash content.

### 2.7. Electrical Conductivity

Electrical conductivity represents a material’s ability to conduct electric current. The electrical conductivity of honey is evaluated by measuring the electrical conductivity of 20 g dry matter of honey in 100 mL distilled water at 20 °C (IHC, 2009). It is expressed in milliSiemens per centimeter (mS/cm).

A previous study showed that electrical conductivity is an indicator of organic acid and mineral content in honey [37]. The organic acid and mineral could dissociate into ions and conduct electricity when in an aqueous solution [38]. In addition to electrical conductivity, ash content is also an indicator of mineral content. Therefore, electrical conductivity is related to ash content [4]. According to IHC, electrical conductivity depends on the ash and acid contents of honey. The higher their content, the higher the electrical conductivity. The IHC recommends that electrical conductivity should not exceed 0.8 mS/cm.

According to the data summarized in Table 7, the electrical conductivity in raw SLBH ranges between 0.26 and 0.604 mS/cm. A previous study reported that SLBH and Manuka honey have a high electrical conductivity of 1.08 and 1.22 mS/cm, respectively, while *Apis* spp. and commercialized honey have electrical conductivity ranges between 0.10 and 0.96 mS/cm [4]. The differences in electrical conductivity levels in honey are contributed by variations in botanical, geographical and entomological differences [4]. Following the dehydration process, the electrical conductivity ranges between 0.25 and 0.563 mS/cm as shown in Table 7. The electrical conductivity before and after the dehydration process was within the permissible limit of not more than 0.8 mS/cm.

Previous studies presented in Table 7 showed that thermal treatment could reduce the electrical conductivity. However, the electrical conductivity could be maintained when subjected to thermal treatment at higher temperature for a short duration. These findings suggest that the dehydration method by thermal treatment can either preserve or reduce the electrical conductivity. Therefore, the temperature at 90 to 95 °C for 15–60 s was the best setting for the thermal treatment method in maintaining the electrical conductivity of SLBH.

Meanwhile, a study showed that the dehydration of SLBH via passive diffusion could increase the electrical conductivity [20]. We observed that it took only three days to increase electrical conductivity to 0.54 mS/cm when SLBH was stored in a clay pot at 35 °C compared to 21 days at 25 °C, as shown in Table 7. These findings suggest that storage of SLBH in a clay pot at a higher temperature setting results in a more rapid increase in electrical conductivity compared to a lower temperature setting. Therefore, storage of SLBH in a clay pot at 35 °C was the better passive diffusion dehydration method.

In conclusion, the dehydration process could decrease, increase, or maintain the electrical conductivity values in SLBH. The dehydration methods that could maintain or increase the electrical conductivity were thermal temperature at high temperature for a short duration and passive diffusion method. By this, the organic acid and mineral contents in honey can be preserved or increased.

### 2.8. Diastase

Diastase is a natural enzyme in honey [9]. However, heating can cause denaturation of the diastase enzyme structure, subsequently lowering the diastase activity [39]. Previously, HMF and diastase activity were parameters for overheating. However, diastase activity is a less reliable parameter compared to HMF because the diastase enzyme level depends on nectar consistency and bee activity [30]. Diastase activity is expressed in diastase number (DN) or Gothe unit (un. Gothe). DN in the Schade scale that corresponds to the Gothe scale number is defined as the amount of starch, measured in grams, hydrolyzed at 40 °C in 1 h per 100 g honey [39]. IHC sets the minimum value of the diastase number as 8 DN. Meanwhile, the Codex Alimentarius Commission sets the minimum value as 3 DN. Previous studies compiled in Table 8 showed that the diastase activity of raw SLBH was between <3–46.1 un. Gothe and between 0 and 0.2 DN. These findings have shown that diastase activity in raw honey varies in nature. After dehydration, the diastase activity ranges between <3–42.7 un. Gothe and 0–0.75 DN as summarized in Table 8.

According to the data presented in Table 8, the dehydration process by thermal treatment could reduce the diastase activity up to 45.1% [15]. Meanwhile, another study by Braghini et al. [16] showed that diastase activity remained unchanged after thermal treatment at 90 to 95 °C for 1 min or less. These findings suggest that dehydration by thermal treatment at a high temperature for a short duration could preserve the diastase activity in SLBH. However, a longer duration dehydration process, even at a lower temperature, reduced the diastase activity. Therefore, thermal treatment at 90 to 95 °C for 15–60 s was the best thermal treatment setting for preserving diastase activity in SLBH.

A study by Yap et al. [17] showed that there was no significant alteration in diastase activity after the dehydration process using a food dehydrator at 40 to 70 °C, as presented in Table 8. These findings suggest that the dehydration method using a food dehydrator is able to maintain the diastase activity, although the dehydration process had a long duration, up to 84 h.

In conclusion, the dehydration process can maintain or decrease diastase activity. The dehydration methods that could maintain diastase activity are thermal treatment at 90 to 95 °C for 15–60 s and the dehydration process using a food dehydrator. The best dehydration method is the food dehydrator method because the diastase activity could be maintained as well as raw SLBH despite long hours of dehydration.

### 2.9. Total Soluble Solids

Soluble solids in honey are sugars, organic acids and minerals. However, honey is predominantly made up of sugar and water. Hence, total soluble solids (TSS) indicate the relation between sugar and water content [40]. The TSS value is moisture dependent. The TSS value increases as the moisture level of honey decreases [41]. Various studies presented in Table 9 showed that the TSS of raw SLBH ranges between 67.9 and 75.5 °Brix. This was consistent with a previous study that reported the TSS of raw SLBH was between 64.5 and 75.8 °Brix [9]. Following the dehydration process, the TSS value ranges between 69 and 86.93 °Brix, as shown in Table 9.

According to the data summarized in Table 9, dehydration by thermal treatment could decrease, increase or maintain TSS. Thermal treatment at low temperature for a long duration or at high temperature for a short duration resulted in more water removal and consequently increased the TSS value.

A study by Yegge et al. [18] showed that the dehydration process using microwave heating and dehumidification methods could increase the TSS values from 67.9 up to 86.93 and 82.69 °Brix, respectively, as presented in Table 9. The microwave heating at 60 PL showed the highest increase in TSS value. These findings were consistent with Table 1, which showed more water reduction at 60 PL compared to 20 and 100 PL. On the other hand, the dehumidification method increased the TSS value, as shown in Table 9 [18]. These findings was consistent with the reduction in water content in SLBH after dehumidification, as presented in Table 1. We observed that microwave heating at 60 PL for 60 s was the best method because it took only 60 s to eliminate the largest amount of water in SLBH, resulting in the highest TSS value.

Previous studies summarized in Table 9 showed that dehydration of SLBH via passive diffusion by clay pot storage could increase the TSS value. A study by Ghazali et al. [19] showed that storage of SLBH in clay pot increased TSS with a higher increase observed in a clay pot with a large surface area. This is consistent with more water reduction reported in a large surface area clay pot compared to a small surface area, as shown in Table 1. Therefore, the dehydration process that eliminated more water resulted in a higher increase in TSS value. On the other hand, another study showed more increase in the TSS value from 72.64 to 73.33 °Brix on day 1 of SLBH storage in a clay pot at 35 °C, compared to 73.33 °Brix at 25 °C storage temperature as shown in Table 9 [20]. These findings suggest that storage in a clay pot at a higher temperature facilitates the dehydration process, resulting in a higher increase in the TSS value.

In conclusion, the TSS value is closely related to moisture content. As the moisture is lost during the dehydration process, the soluble solids in SLBH become more concentrated, thus, increasing the TSS value. The methods that could increase TSS value are thermal treatment, microwave heating, dehumidification and passive diffusion via clay pot storage. Microwave heating at 60 PL is the best dehydration setting with regards to TSS because it resulted in the highest increase in the TSS level in a short period.

### 2.10. Total Reducing Sugar

Fructose and glucose are the primary reducing sugars in honey [37]. Total reducing sugar is the sum of fructose and glucose. According to the IHC (2009) guideline, a good quality honey should have total reducing sugars of at least 60 g/100 g. Several studies summarized in Table 10 showed that fructose content in raw SLBH ranges between 9.4 and 39.4 g/100 g, glucose content between 3.41 and 22.8 g/100 g, and total reducing sugar between 15.8 and 59.4 g/100 g. After the dehydration process, the fructose content ranges between 8.2 and 44.6 g/100 g, glucose content between 3.46 and 25 g/100 g, and total reducing sugar between 18.2 and 68.1 g/100 g.

According to the data presented in Table 10, thermal treatment at 52 to 71 °C increased both fructose and glucose content [15]. Therefore, the total reducing sugars increased up to 69.6 g/100 g, which was within the IHC standard of not less than 60 g/100 g [15]. Meanwhile, another study showed that thermal treatment at a higher temperature of 90 to 95 °C for 15–60 s decreased fructose but increased the glucose content [16]. As a result, the total reducing sugar decreased to a range between 55.1 and 55.7 g/100 g, which was less than the IHC standard [16]. These findings suggest that thermal treatment can increase or decrease the reducing sugar content in SLBH.

A study by Chen et al. [21] measured the level of reducing sugar before and after the dehydration process using vacuum and freeze-drying at a 5% moisture setting. As shown in Table 10, vacuum drying decreased the fructose content. Meanwhile, the glucose content was reduced from in vacuum drying at 40–50 °C but increased at 60 °C. As a result, the total reducing sugar was decreased in vacuum drying at 40–50 °C, and increased at 60 °C. On the other hand, the freeze-drying method decreased the fructose and glucose content, and consequently, the total reducing sugar decreased. It is observed that vacuum drying at 60 °C was the only method that could increase the total reducing sugar in SLBH, even though the value was still below the standard of at least 60 g/100 g. Therefore, vacuum drying at a higher temperature at 60 °C was the best method compared to vacuum drying at 40–50 °C and freeze-drying.

From the data provided in Table 10, the dehydration process by the MARDI dehydrator could increase both fructose and glucose content [24]. As a result, the total reducing sugar was increased from 15.8 to 29.32 g/100 g. Although the total reducing sugar was below 60 g/100 g after dehydration process, the MARDI dehydrator could increase the total reducing sugars in SLBH.

A previous study showed that dehydration of SLBH via passive diffusion by storage in a clay pot with different surface areas at room temperature of 25 °C for 10 days altered the fructose and glucose content as presented in Table 10 [19]. The fructose content was reduced when stored in a clay pot with a smaller surface area, while it increased in a clay pot with a larger surface area. The glucose content increased in both clay pots with larger and smaller surface area. The total reducing sugar increased in both clay pot with larger and smaller surface area, but the values were less than 60 g/100 g. Meanwhile, storage of SLBH in a glass container after 10 days reduced fructose, glucose and total reducing sugar [19]. Therefore, dehydration via passive diffusion could increase the total reducing sugar compared to a glass container storage. These findings suggest that storage in a clay pot promotes water loss, especially in a clay pot with a larger surface area. Consequently, the fructose and glucose concentration will increase. Therefore, dehydration via passive diffusion was the best with a clay pot with a large surface area.

In conclusion, the dehydration process will alter the fructose, glucose and total reducing sugar in SLBH. The dehydration methods that could increase the total reducing sugar are thermal treatment at 52 to 71 °C for 470 min to 24 s, vacuum drying at 60 °C, MARDI dehydrator and passive diffusion in a clay pot with a larger surface area. However, passive diffusion took up to 10 days to increase the total reducing sugar. Meanwhile, the duration of vacuum drying was not mentioned by the authors.

### 2.11. Total Phenolic Content

The phenolic content (TPC) is a strong indicator of antioxidants in honey [1]. TPC strongly correlates with antioxidant components, which are ferric reducing antioxidant power (FRAP) and *β*-carotene bleaching inhibition [42]. Therefore, the presence of phenolics is a sign of a good quality SLBH. Exposure of honey to heat will release the phenolic components in the honey, subsequently increasing the total phenolic content [43]. On the other hand, heat can trigger the Maillard reaction in honey. As a result, Maillard reaction products (MRPs) such as brown melaidonins will be released and increase the TPC value [22]. TPC is expressed as milligram of gallic acid equivalents per gram of honey (mg GAE/g), per 100 g of honey (mg GAE/100 g) or per kilogram of honey (mg GAE/kg). This review standardized the TPC value in mg GAE/100 g. According to the data summarized in Table 11, the TPC of raw SLBH was between 12.45 and 5130 mg GAE/100 g. A previous study showed that SLBH has the highest TPC level before and after the dehydration process compared to Tualang and Acacia honey [23]. Following the dehydration process, the TPC level ranges between 11.05 and 6750 mg GAE/100 g as presented in Table 11.

Several studies summarized in Table 11 showed that thermal treatment could increase the TPC level. Another study by Chong et al. [22] showed that both thermosonication and thermal treatment could increase the TPC level. However, the thermosonication method has increased the TPC level of honey up to 58%, and up to 54% using the thermal treatment. These findings suggest that thermosonication is a better dehydration method than thermal treatment in raising the TPC level of SLBH.

A study by Chen et al. [21] showed that vacuum treatment and freeze-drying methods could increase the TPC level. In the study, vacuum drying at a 5% moisture setting showed the highest increase in TPC level as shown in Table 11. On the other hand, an increase in TPC level was observed together with an increase in the dehydration temperature. Vacuum drying with 5% moisture at 50–60 °C could increase the TPC level up to 35 mg GAE/100 g compared to 25 mg GAE/100 g at 40 °C. These findings suggest that vacuum drying with a 5% moisture content set at 50–60 °C is the best method compared to vacuum drying and evaporation at an 11% moisture level, and the freeze-drying method.

From the data provided in Table 11, dehydration of SLBH using microwave heating and dehumidification could either decrease or increase the TPC level. The microwave heating method at 60 PL for 60 s showed the highest increase in TPC compared to other temperature and duration settings [18]. On the other hand, the dehumidification process increased the TPC level after two days of treatment [18]. These findings suggest that microwave heating at 60 PL for 60 s is the better method because it took less time, 1 min, to increase more TPC compared to the dehumidification process, which took up to two days.

A study by Yap et al. [17] showed dehydration of SLBH using a food dehydrator for 12–84 h could increase the TPC value, as presented in Table 11. The TPC value increased from 41.99 to 57.83, 73.77 and 157.32 mg GAE/100 g at 40, 50 and 70 °C, respectively. Meanwhile, another study showed that dehydration using the MARDI dehydrator at a lower temperature of 30 °C for 8 h increased the TPC value from 24.47 to 25 mg GAE/100 g [24]. These findings suggest that the higher the dehydrator temperature, the higher the increase in the TPC value.

Various dehydration methods can increase the TPC value in SLBH. The dehydration methods that could increase TPC level are thermal treatment, thermosonication, vacuum treatment, freeze-drying, microwave heating, dehumidification and food and MARDI dehydrators. We observed that microwave heating at 60 PL is the best dehydration method because it could increase the TPC within a 60 s duration. However, the duration of the vacuum method was not mentioned by the authors.

### 2.12. Total Flavonoid Content

The flavonoid content (TFC) is an indicator of antioxidants in honey [25]. TFC has a strong correlation with biochemical antioxidant indicators, such as DPPH free radical scavenging activity, ferric reducing antioxidant power (FRAP) and *β*-carotene bleaching inhibition [42]. When honey is heated, the flavonoids are released from their bonds, thus increasing the TFC value. Flavonoids, such as quercetin, catechin and rutin, are commonly used as the reference standard. For example, if quercetin is used as a reference standard, it can be expressed as a milligram of quercetin equivalent per gram of honey (mg QE/g) or per kilogram of honey (mg QE/kg). Several studies summarized in Table 12 showed that the TFC levels of raw SLBH were between 0.2 and 32.3 mg QE/g. The TFC level after the dehydration process was increased to between 0.2 and 36.43 mg QE/g, as shown in Table 12. A previous study showed that SLBH has the highest TFC level before and after the dehydration process compared to Tualang and Acacia honey [23].

According to the data presented in Table 12, a study by Sulaiman and Sarbon [23] showed that thermal treatment at 50 to 90 °C increased the TFC level as the dehydration temperature increased. Therefore, the highest thermal treatment temperature at 90 °C was the best setting to increase the TFC level. However, the duration was not mentioned by the author. Meanwhile, another study by Chen et al. [21] showed that the dehydration process using vacuum and freeze-drying methods could increase the TFC level. At a 5% moisture setting, vacuum drying could increase more TFC compared to an 11% moisture setting. In addition to that, a greater increase in TFC level was observed when the vacuum drying was set at a lower moisture setting of 5% compared to the freeze-drying method [21]. These findings suggest that the vacuum drying method with low moisture content at 5% moisture setting is the best method to increase the TFC level.

In conclusion, all dehydration methods could increase the TFC level in SLBH. A greater increase in TFC level was seen when SLBH was dehydrated to a low moisture content level and at a high temperature. The dehydration methods that could increase TFC levels are thermal treatment at 90 °C and vacuum drying at a 5% moisture level. However, the duration of both methods was not mentioned by the authors. Therefore, we suggest the dehydration method be performed at a high temperature that can eliminate as much water as possible to elevate both TFC and antioxidant levels.

### 2.13. Individual Phenolic Compounds

Table 13 and Table 14 summarized the quantified individual phenolic compounds in raw and dehydrated SLBH comprised of chlorogenic acid, rosmarinic acid, rutin and quercetin. Thermal treatment increased the chlorogenic acid, rutin and quercetin in SLBH. However, the amount was higher at a lower heat intensity. Dehydration at 90 and 95 °C for 15 s resulted in a higher increase in chlorogenic acid level compared to 60 s duration, as shown in Table 13. Furthermore, a greater increase in rutin and quercetin values was seen at 60 °C for 22 min compared to higher temperatures or extended duration settings, as presented in Table 14. We observed that increased heat exposure of SLBH resulted in a lesser increase in phenolic compounds. This scenario may be due to the heat-transformation of phenolic compounds. A previous study showed the transformation of rutin (glycones) to isoquercitrin (aglycones) when subjected to thermal processing [44]. These findings suggest the biochemical events that happened during the heat processing of raw SLBH may have led to the formation of new phenolic compounds.

According to the data compiled in Table 13 and Table 14, the vacuum drying increased all the phenolic compounds except rutin. A higher value of phenolics was observed as the temperature of vacuum drying increased. On the other hand, freeze-drying increased all the individual phenolic compounds. Wang et al. [45] explained that some phenolic compounds were released during heat processing, leading to an increase in phenolics. Another study showed that more phenolic compounds are released at higher treatment temperatures [46].

In conclusion, the dehydration process could increase the phenolic compounds in SLBH. The heat from dehydration may increase the value of the phenolic compounds present in SLBH due to their liberation during processing. However, prolonged heat exposure resulted in lesser increase in phenolic compounds due to the possibility of their conversion to other compounds.

### 2.14. The Optimal Setting for Each Method of Dehydration

Table 15 summarized the optimal temperature and duration for each dehydration method used in previous studies. According to the data presented in Table 15, the most optimal dehydration processes for SLBH are thermal treatment at 90–95 °C for 15–60 s, thermal treatment at 45–90 °C for 30–120 min, thermosonication at 45–90 °C for 30–120 min, vacuum drying at 5% moisture content and 60 °C, freeze-drying at 5% moisture content and −54 °C, microwave heating at 60 PL for 60 s, dehumidification at 35 °C for two days, food dehydrator at 55 °C for 18 h, MARDI dehydrator at 30 °C for 8 h, dehydration by passive diffusion by storage in a clay pot with a large surface area at 25 °C for 10 days and storage in a clay pot at 35 °C for three days.

A suitable dehydration method is a method that can reduce the moisture content to below 17% and water activity to less than 0.6. These conditions prevent the fermentation process and microorganism growth. At the same time, the HMF content must be below the permitted level of 40 mg/kg. The pH level should be increased or maintained. Hence, the SLBH will not be too acidic, and the sourness can be prevented. Meanwhile, the free acidity, ash content, electrical conductivity and diastase activity should be increased or maintained. As a result, honey’s organic acids, enzymes, and minerals can be improved or preserved as well as in fresh SLBH. The water loss in SLBH through the dehydration process is reflected by an increase in the total soluble solids and total reducing sugar. On the other hand, a good dehydration method will increase the antioxidant activity in SLBH by increasing the total phenolic and flavonoid content.

Table 15 shows that vacuum drying at 5% moisture content and 60 °C, freeze-drying at 5% moisture content and −54 °C for 24 h, and food dehydrator at 55 °C for 18 h could extract 80% and more water content in SLBH. As a result, these methods could decrease both moisture content below 17% and water activity to less than 0.6. The HMF value remain within the permissible range of below 40 mg/kg. Microwave heating at 60 PL for 60 s could reduce moisture below 17%. However, there was a lack of data on water activity and HMF content. On the other hand, the total phenolic content increased after dehydration by these methods.

## 3. Conclusions

Regardless of the dehydration method used, it was observed that the dehydration process at a high temperature resulted in a greater moisture content reduction. However, a very high temperature and prolonged honey exposure to extreme heat can increase the undesirable HMF content. Therefore, the dehydration process should be performed at an optimal temperature that can extract the maximum amount of water feasible while maintaining a low HMF level within the permitted amount. This review compiles data on dehydration of SLBH by thermal treatment, thermosonication, vacuum method, freeze-drying, microwave heating, dehumidification, dehydration using the MARDI dehydrator and dehydration via passive diffusion by a clay pot. This review found that the dehydration process using vacuum drying at 5% moisture content and 60 °C, freeze-drying at 5% moisture content and −54 °C for 24 h, and food dehydrator at 55 °C for 18 h could remove 80% and more water content in SLBH. As a result, these methods could decrease moisture content below 17% and water activity to less than 0.6. The HMF values were within the permissible range set by Codex Alimentarius Standards (2001) of below 40 mg/kg. The total phenolic content increased after dehydration by these methods. The physicochemical parameters of dehydrated SLBH are not comprehensive. Therefore, we suggest that future studies on dehydration of SLBH include moisture content, water activity, HMF, pH, free acidity, ash content, electrical conductivity, diastase activity, total soluble solids, total reducing sugar, total phenolic content and total flavonoid content as the parameters. Furthermore, we suggest more studies to evaluate phenolic compounds before and after the dehydration of SLBH. By this, we can compare and choose the best dehydration method to maximize the nutritional benefits of SLBH.

## Figures and Tables

**Table 1 molecules-27-07243-t001:** Moisture content of raw and dehydrated stingless bee honey (SLBH).

Method of Dehydration		SLBH Species	T (°C)/ PL	Time	Moisture Content (%)		Water Reduction (%)	Author
					Raw	Dehydrated		
Thermal treatment		*Tetragonisca angustula*	52 °C	470 min	23.9	21.6 *	9.6	[15]
		*T. angustula*	55 °C	170 min	23.9	22.7 *	5.0	
		*T. angustula*	57 °C	60 min	23.9	23.3 *	2.5	
		*T. angustula*	60 °C	22 min	23.9	23.3 *	2.5	
		*T. angustula*	66 °C	8 min	23.9	23.3 *	2.5	
		*T. angustula*	66 °C	3 min	23.9	23.4	2.1	
		*T. angustula*	68 °C	1 min	23.9	23.4	2.1	
		*T. angustula*	71 °C	24 s	23.9	23.5	1.7	
		*Melipona bicolor*	90 °C	15–60 s	30.8	29.5 *	4.2	[16]
		*M. bicolor*	95 °C	15–60 s	30.8	29.5–29.6 *	3.9–4.2	
		-	45–90 °C	30–120 min	30.93	28.8	6.9	[22]
Thermosonication		-	45–90 °C	30–120 min	31.06	25.9	16.6	
Vacuum	Drying (5% moisture)	*Heterotrigona itama*	40–60 °C	-	31.9	5	84.3	[21]
	Drying (11% moisture)	*H. itama*	40–60 °C	-	31.9	11	65.5	
	Evaporation (11% moisture)	*H. itama*	40–60 °C	-	31.9	11	65.5	
Freeze-drying (5% moisture)		*H. itama*	−54 °C	24 h	31.9	5	84.3	
Microwave heating		*H. itama*	20 PL	15–60 s	31.47	25.24–26.46	16–20	[18]
		*H. itama*	60 PL	25–60 s	31.47	15.04–20.3 *	35–52	
		*H. itama*	100 PL	5–15 s	31.47	22.29–24.32 *	23–29	
Dehumidification		*H. itama*	35 °C	1–2 days	31.47	17.21–17.48 *	44–45	
Food dehydrator		*H. itama*	40 °C	36 h	40	<8	>80	[17]
		*H. itama*	55 °C	18 h	40	<8	>80	
		*H. itama*	55 °C	36 h	40	0	100	
		*H. itama*	70 °C	18 h	40	<8	>80	
		*H. itama*	70 °C	36 h	40	0	100	
MARDI dehydrator		*H. itama*	30 °C	8 h	29	19	35	[24]
Glass bottle storage		*Geniotrigona thoracica*	25 °C	1–10 days	26.21	25–26	0.8–4.62	[19]
Clay pot storage	Small surface area	*G. thoracica*	25 °C	1–10 days	26.21	24.32 *	7.21	
	Large surface area	*G. thoracica*	25 °C	1–10 days	26.21	23.35 *	10.9	
		*H. itama*	25 °C	1–21 days	25.82	18.13–25.13 *	2.7–29.8	[20]
		*H. itama*	35 °C	1–3 days	25.82	19.56–23.68 *	8.3–24.2	

* a significant difference (*p* < 0.05) compared to raw SLBH; T: temperature; PL: power level.

**Table 2 molecules-27-07243-t002:** Water activity of raw and dehydrated stingless bee honey (SLBH).

Method of Dehydration		SLBH Species	T (°C)	Time	Water Activity		Water Activity Reduction (%)	Author
					Raw	Dehydrated		
Thermal treatment		-	45–80	30–100 min	0.795	<0.767	<3.5	[22]
		-	90	120 min	0.795	0.767	3.5	
Thermosonication		-	45–80	30–100 min	0.807	<0.743	<7.9	
		-	90	120 min	0.807	0.743	7.9	
Vacuum	Drying (5% moisture)	*Heterotrigona itama*	40–60	-	0.79	0.28–0.29 *	63.3–64.6	[21]
	Drying (11% moisture)	*H. itama*	40–60	-	0.79	0.45–0.48 *	39.2–43	
	Evaporation (11% moisture)	*H. itama*	40–60	-	0.79	0.47–0.5 *	36.7–40.5	
Freeze-drying (5% moisture)		*H. itama*	−54	24 h	0.79	0.3 *	62	
Food dehydrator		*H. itama*	40	36 h	0.788	<0.6	>23.9	[17]
		*H. itama*	55	18 h	0.788	<0.6	>23.9	
		*H. itama*	70	12 h	0.788	<0.6	>23.9	
Glass bottle storage		*Geniotrigona thoracica*	25	1–10 days	0.8	0.782–0.785	1.9–2.25	[19]
Clay pot storage	Small surface area	*G. thoracica*	25	1–10 days	0.8	0.679–0.774 *	3.3–15.1	
	Large surface area	*G. thoracica*	25	1–10 days	0.8	0.632–0.737 *	7.9–21	
		*H. itama*	25	1 day	0.79	0.79	-	[20]
		*H. itama*	25	7–21 days	0.79	0.63–0.7 *	11.4–20.3	
		*H. itama*	35	1–3 days	0.79	0.7–0.76 *	3.8–11.4	

* a significant difference (*p* < 0.05) compared to raw SLBH; T: temperature.

**Table 3 molecules-27-07243-t003:** Hydroxymethylfurfural (HMF) content of raw and dehydrated stingless bee honey (SLBH).

Method of Dehydration	SLBH Species	T (°C)	Time	HMF (mg/kg)		Author
				Raw	Dehydrated	
Thermal treatment	*Tetragonisca angustula*	52	470 min	<LOQ	<LOQ	[15]
	*T. angustula*	55	170 min	<LOQ	<LOQ	
	*T. angustula*	57	60 min	<LOQ	<LOQ	
	*T. angustula*	60	22 min	<LOQ	<LOQ	
	*T. angustula*	66	8 min	<LOQ	<LOQ	
	*T. angustula*	66	3 min	<LOQ	<LOQ	
	*T. angustula*	68	1 min	<LOQ	<LOQ	
	*T. angustula*	71	24 s	<LOQ	<LOQ	
	*Melipona bicolor*	90	15–60 s	<LOQ	<LOQ	[16]
	*M. bicolor*	95	15–60 s	<LOQ	<LOQ	
	-	75–95	20–60 s	<LOQ	<LOQ	[32]
	-	75	15 min	<LOQ	<LOQ	
	-	75	24 h	<LOQ	238.18	
	-	45–67.5	30–75 min	0	0	
	-	67.5–90	100–120 min	0	↑ up to 42.40 *	[22]
Thermosonication	-	45–67.5	30–75 min	0	0	
	-	67.5–90	100–120 min	0	↑ up to 62.46 *	
Vacuum drying (5% moisture)	*Heterotrigona itama*	40	-	0	9.3 *	[21]
	*H. itama*	50	-	0	10.71 *	
	*H. itama*	60	-	0	12.18 *	
Freeze-drying (5% moisture)	*H. itama*	−54	-	0	9.29 *	
Food dehydrator	*H. itama*	40	18–36 h	0	0	[17]
	*H. itama*	55	18 h	0	<5.81	
	*H. itama*	55	36 h	0	5.81	
	*H. itama*	70	18 h	0	<50	
	*H. itama*	70	36 h	0	83.19	
MARDI dehydrator	*H. itama*	30	8 h	2.27	2.39	[24]

* a significant difference (*p* < 0.05) compared to raw SLBH; T: temperature; LOQ: limit of quantification; ↑: increase.

**Table 4 molecules-27-07243-t004:** pH of raw and dehydrated stingless bee honey (SLBH).

Method of Dehydration		SLBH Species	T (°C)/ PL	Time	pH		Alteration (%)	Author
					Raw	Dehydrated		
Thermal treatment		*Tetragonisca angulusta*	52 °C	470 min	5.18	4.93 *	↓ 4.8	[15]
		*T. angulusta*	55 °C	170 min	5.18	5 *	↓ 3.5	
		*T. angulusta*	57 °C	60 min	5.18	5.04	Not sig.	
		*T. angulusta*	60 °C	22 min	5.18	5.01 *	↓ 3.3	
		*T. angulusta*	66 °C	8 min	5.18	5.08	Not sig.	
		*T. angulusta*	66 °C	3 min	5.18	5.01 *	↓ 3.3	
		*T. angulusta*	68 °C	1 min	5.18	5.03	Not sig.	
		*T. angulusta*	71 °C	24 s	5.18	5.14	Not sig.	
		*Melipona bicolor*	90 °C	15–60 s	3.25	3.25	–	[16]
		*M. bicolor*	95 °C	15–60 s	3.25	3.26	Not sig.	
		-	50 °C	-	3.81	3.85	Not sig.	[23]
		-	75 °C	-	3.81	3.89 *	↑ 2.1	
		-	90 °C	-	3.81	3.96 *	↑ 3.9	
Vacuum	Drying (5% moisture)	*Heterotrigona itama*	40–60 °C	-	3.16	3.36–3.41 *	↑ 6.3–7.9	[21]
	Drying (11% moisture)	*H. itama*	40–60 °C	-	3.16	3.21–3.31 *	↑ 1.6–4.7	
	Evaporation (11% moisture)	*H. itama*	40–60 °C	-	3.16	3.2–3.29	Not sig.	
Freeze-drying (5% moisture)		*H. itama*	−54 °C	-	3.16	3.14	Not sig.	
Microwave heating		*H. itama*	20 PL	15 s	3.58	3.5	Not sig.	[18]
		*H. itama*	20 PL	30–60 s	3.58	3.45–3.48 *	↓ 2.8–3.6	
		*H. itama*	60 PL	25–30 s	3.58	3.45 *	↓ 3.6	
		*H. itama*	60 PL	60 s	3.58	3.51	Not sig.	
		*H. itama*	100 PL	5–15 s	3.58	3.46–3.47 *	↓ 3.1–3.4	
Dehumidification		*H. itama*	35 °C	1 day	3.58	3.54 *	↓ 1.1	
		*H. itama*	35 °C	2 days	3.58	3.62	Not sig.	
Glass bottle storage		*Geniotrigona thoracica*	25 °C	1–10 days	3.11	3.11–3.13	Not sig.	[19]
Clay pot storage	Small surface area	*G. thoracica*	25 °C	1–10 days	3.11	3.12–3.16	Not sig.	
	Large surface area	*G. thoracica*	25 °C	1–10 days	3.11	3.11	–	
		*H. itama*	25 °C	1–7 days	3.44	3.55–3.62 *	↑ 3.2–5.2	[20]
		*H. itama*	25 °C	14–21 days	3.44	3.34–3.38 *	↓ 1.7–2.9	
		*H. itama*	35 °C	1–3 days	3.44	3.52–3.63 *	↑ 2.3–5.5	

* a significant difference (*p* < 0.05) compared to raw SLBH; Not sig.: no significant difference (*p* > 0.05) compared to raw SLBH; T: temperature; PL: power level; ↑: increase; ↓: decrease.

**Table 5 molecules-27-07243-t005:** Free acidity of raw and dehydrated stingless bee honey (SLBH).

Method of Dehydration		SLBH Species	T (°C)	Time	Free Acidity (meq/kg)		Alteration (%)	Author
					Raw	Dehydrated		
Thermal treatment		*Tetragonisca angulusta*	52	470 min	23.2	23.3	Not sig.	[15]
		*T. angulusta*	55	170 min	23.2	21.7	Not sig.	
		*T. angulusta*	57	60 min	23.2	21.3 *	↓ 8.2	
		*T. angulusta*	60	22 min	23.2	21.4 *	↓ 7.8	
		*T. angulusta*	66	8 min	23.2	21.4 *	↓ 7.8	
		*T. angulusta*	66	3 min	23.2	21.3 *	↓ 8.2	
		*T. angulusta*	68	1 min	23.2	23	Not sig.	
		*T. angulusta*	71	24 s	23.2	21.3 *	↓ 8.2	
		*Melipona bicolor*	90	15–60 s	32.9	31.4–31.5	Not sig.	[16]
		*M. bicolor*	95	15 s	32.9	32.2	Not sig.	
		*M. bicolor*	95	60 s	32.9	33.5	Not sig.	
Vacuum	Drying (5% moisture)	*Heterotrigona itama*	40–60	-	152.5	112.5–117 *	↓ 23.3–26.2	[21]
	Drying (11% moisture)	*H. itama*	40–60	-	152.5	120–132 *	↓ 13.4–21.3	
	Evaporation (11% moisture)	*H. itama*	40–60	-	152.5	113–123.5 *	↓ 19–25.9	
Freeze-drying (5% moisture)		*H. itama*	−54	-	152.5	150.5	Not sig.	
Glass bottle storage		*Geniotrigona thoracica*	25	1–10 days	172	174–177	Not sig.	[19]
Clay pot storage	Small surface area	*G. thoracica*	25	1–10 days	172	179–181	Not sig.	
	Large surface area	*G. thoracica*	25	1–10 days	172	174–178	Not sig.	
		*H. itama*	25	1–7 days	85	88–99 *	↑ 3.5–16.5	[20]
		*H. itama*	25	14–21 days	85	82–83 *	↓ 2.4–3.5	
		*H. itama*	35	1–3 days	85	89–94	Not sig.	

* a significant difference (*p* < 0.05) compared to raw SLBH; Not sig.: no significant difference (*p* > 0.05) compared to raw SLBH; T: temperature; ↑: increase; ↓: decrease.

**Table 6 molecules-27-07243-t006:** Ash content of raw and dehydrated stingless bee honey (SLBH).

Method of Dehydration		SLBH Species	T (°C)	Duration (days)	Ash (g/100 g)		Alteration (%)	Author
					Raw	Dehydrated		
Glass bottle storage		*Geniotrigona thoracica*	25	1–10	0.055	0.049–0.057	Not sig.	[19]
Clay pot storage	Small surface area	*G. thoracica*	25	1–7	0.055	0.051–0.073	Not sig.	
		*G. thoracica*	25	10	0.055	0.088 *	↑ 60	
	Large surface area	*G. thoracica*	25	1–4	0.055	0.059–0.081	Not sig.	
		*G. thoracica*	25	7–10	0.055	0.091–0.092 *	↑ 65.5–67.3	
		*Heterotrigona itama*	25	1–21	0.11	0.11–0.18	Not sig.	[20]
		*H. itama*	35	1–3	0.11	0.1–0.15	Not sig.	

* a significant difference (*p* < 0.05) compared to raw SLBH; Not sig.: no significant difference (*p* > 0.05) compared to raw SLBH; T: temperature; ↑: increase.

**Table 7 molecules-27-07243-t007:** Electrical conductivity of raw and dehydrated stingless bee honey (SLBH).

Method of Dehydration	SLBH Species	T (°C)	Time	Electrical Conductivity (mS/cm)		Alteration		Author
				Raw	Dehydrated			
Thermal treatment	*Tetragonisca angustula*	52	470 min	0.604	0.514 *		↓	[15]
	*T. angustula*	55	170 min	0.604	0.545 *		↓	
	*T. angustula*	57	60 min	0.604	0.53 *		↓	
	*T. angustula*	60	22 min	0.604	0.525 *		↓	
	*T. angustula*	66	8 min	0.604	0.517 *		↓	
	*T. angustula*	66	3 min	0.604	0.563 *		↓	
	*T. angustula*	68	1 min	0.604	0.56 *		↓	
	*T. angustula*	71	24 s	0.604	0.531 *		↓	
	*Melipona bicolor*	90	15–60 s	0.26	0.26		-	[16]
	*M. bicolor*	95	15–60 s	0.26	0.25–0.26		Not sig.	
Clay pot storage	*H. itama*	25	1–21 days	0.43	0.49–0.54 *		↑	[20]
	*H. itama*	35	1–3 days	0.43	0.51–0.54 *		↑	

* a significant difference (*p* < 0.05) compared to raw SLBH; Not sig.: no significant difference (*p* > 0.05) compared to raw SLBH; T: temperature; ↑: increase; ↓: decrease.

**Table 8 molecules-27-07243-t008:** Diastase activity of raw and dehydrated stingless bee honey (SLBH).

Method of Dehydration	SLBH Species	T (°C)	Time	Diastase ^a^ (un. Gothe) ^b^ (Diastase Number)		Alteration (%)	Author
				Raw	Dehydrated		
Thermal treatment	*Tetragonisca angustula*	52	470 min	46.1 ^a^	35.3 ^a,^*	↓ 23.4	[15]
	*T. angustula*	55	170 min	46.1 ^a^	30.3 ^a,^*	↓ 34.3	
	*T. angustula*	57	60 min	46.1 ^a^	31.9 ^a,^*	↓ 30.8	
	*T. angustula*	60	22 min	46.1 ^a^	32.4 ^a,^*	↓ 29.7	
	*T. angustula*	66	8 min	46.1 ^a^	27.6 ^a,^*	↓ 40.1	
	*T. angustula*	66	3 min	46.1 ^a^	29.3 ^a,^*	↓ 36.4	
	*T. angustula*	68	1 min	46.1 ^a^	25.3 ^a,^*	↓ 45.1	
	*T. angustula*	71	24 s	46.1 ^a^	42.7 ^a,^*	↓ 7.4	
	*Melipona bicolor*	90	15–60 s	<3 ^a^	<3 ^a^	-	[16]
	*M. bicolor*	95	15–60 s	<3 ^a^	<3 ^a^	-	
Food dehydrator	*Heterotrigona itama*	40	12–84 h	0–0.2 ^b^	0–0.6 ^b^	Not sig.	[17]
	*H. itama*	55	12–84 h	0–0.2 ^b^	0–0.2 ^b^	Not sig.	
	*H. itama*	70	12–84 h	0–0.2 ^b^	0–0.75 ^b^	Not sig.	

* a significant difference (*p* < 0.05) compared to raw SLBH; ^a^: the value expressed in Gothe unit (un. Gothe); ^b^: value expressed in Diastase Number (DN); Not sig.: no significant difference (*p* > 0.05) compared to raw SLBH; T: temperature; ↓: decrease.

**Table 9 molecules-27-07243-t009:** Total soluble solid of raw and dehydrated stingless bee honey (SLBH).

Method of Dehydration		SLBH Species	T (°C)/ PL	Time	Total Soluble Solids (°Brix)		Alteration (%)	Author
					Raw	Dehydrated		
Thermal treatment		*Tetragonisca angustula*	52 °C	470 min	75.5	77.7 *	↑ 2.9	[15]
		*T. angustula*	55 °C	170 min	75.5	76.5 *	↑ 1.3	
		*T. angustula*	57 °C	60 min	75.5	75.8	↑ 0.004	
		*T. angustula*	60 °C	22 min	75.5	75.7	↑ 0.3	
		*T. angustula*	66 °C	8 min	75.5	74 *	↓ 2.0	
		*T. angustula*	66 °C	3 min	75.5	75.5	-	
		*T. angustula*	68 °C	1 min	75.5	75.5	-	
		*T. angustula*	71 °C	24 s	75.5	75.5	-	
		*Melipona bicolor*	90 °C	15–60 s	68.5	69–69.4	↑ 0.7–1.3	[16]
		*M. bicolor*	95 °C	15–60 s	68.5	69.4	↑ 1.31	
Microwave heating		*Heterotrigona itama*	20 PL	15–60 s	67.9	72.2–73.43	↑ 6.3–8.1	[18]
		*H. itama*	60 PL	25–60 s	67.9	75.4–86.93 *	↑ 10.5–28	
		*H. itama*	100 PL	5–15 s	67.9	72.73–76.33 *	↑ 7.1–12.4	
Dehumidification		*H. itama*	35 °C	1–2 days	67.9	82.52–82.69 *	↑ 21.5–21.8	
Glass bottle storage		*Geniotrigona thoracica*	25 °C	1–4 days	72.2	71.7–72.0	↓ 0.3–0.7	[19]
		*G. thoracica*	25 °C	7–10 days	72.2	72.5–72.9	↑ 0.3–1.0	
Clay pot storage	Small surface area	*G. thoracica*	25 °C	1–10 days	72.2	73.3–79.3 *	↑ 1.5–9.8	
	Large surface area	*G. thoracica*	25 °C	1–10 days	72.2	77–82.9 *	↑ 6.6–14.8	
		*H. itama*	25 °C	1–21 days	72.64	73.33–80.25 *	↑ 0.9–10.5	[20]
		*H. itama*	35 °C	1–3 days	72.64	74.75–78.85 *	↑ 2.9–8.5	

* a significant difference (*p* < 0.05) compared to raw SLBH; T: temperature; PL: power level; ↑: increase; ↓: decrease.

**Table 10 molecules-27-07243-t010:** Sugar content of raw and dehydrated stingless bee honey (SLBH).

Method of Dehydration		SLBH Species	T (°C)	Time	Fructose (g/100 g)		Glucose (g/100 g)		Total Reducing Sugar (g/100 g)		Author
					Raw	Dehydrated	Raw	Dehydrated	Raw	Dehydrated	
Thermal treatment		*Tetragonisca angulusta*	52	470 min	39.4	42.5 *	20	23 *	59.4	65.5	[15]
		*T. angulusta*	55	170 min	39.4	42.2 *	20	23.5 *	59.4	65.7	
		*T. angulusta*	57	60 min	39.4	39.5	20	21.9 *	59.4	61.4	
		*T. angulusta*	60	22 min	39.4	42.9 *	20	24.2 *	59.4	67.1	
		*T. angulusta*	66	8 min	39.4	44.6 *	20	25 *	59.4	69.6	
		*T. angulusta*	66	3 min	39.4	40.6	20	21.8 *	59.4	62.4	
		*T. angulusta*	68	1 min	39.4	43.9 *	20	24.2 *	59.4	68.1	
		*T. angulusta*	71	24 s	39.4	40.9	20	23.8 *	59.4	64.7	
		*Melipona bicolor*	90	15–60 s	33.9	31.8–31.9 *	22.8	23.3–23.7	56.7	55.1–55.6	[16]
		*M. bicolor*	95	15–60 s	33.9	31.5–31.9 *	22.8	23.6–23.8	56.7	55.1–55.7	
Vacuum drying (5% moisture)		*Heterotrigona itama*	40	-	10.09	9.37	14.86	13.68 *	24.95	23.05	[21]
		*H. itama*	50	-	10.09	9.19	14.86	14.51	24.95	23.7	
		*H. itama*	60		10.09	9.79	14.86	15.45	24.95	25.24	
Freeze-drying (5% moisture)		*H. itama*	−54	-	10.09	9.94	14.86	14.1	24.95	24.04	
MARDI dehydrator		*H. itama*	30	8 h	12.39	13.52	3.41	3.46	15.8	29.32	[24]
Glass bottle storage		*Geniotrigona thoracica*	25	1–10 days	9.4–9.5	8.2–9 *	11	10.0–11.0 *	20.4–20.5	18.2–20	[19]
Clay pot storage	Small surface area	*G. thoracica*	25	1 day	9.4–9.5	8.6 *	11	10.0–11.0	20.4–20.5	18.6–19.6	
		*G. thoracica*	25	4 days	9.4–9.5	8.9 *	11	11.0–12.0 *	20.4–20.5	19.9–20.9	
		*G. thoracica*	25	7 days	9.4–9.5	8.9–9.0 *	11	11.0–12.0 *	20.4–20.5	19.9–21	
		*G. thoracica*	25	10 days	9.4–9.5	9.3 *	11	12 *	20.4–20.5	21.3	
	Large surface area	*G. thoracica*	25	1–7 days	9.4–9.5	9.2–9.45 *	11	11.0–12.0 *	20.4–20.5	20.2–21.45	
		*G. thoracica*	25	10 days	9.4–9.5	9.7 *	11	13.0 *	20.4–20.5	22.7	

* a significant difference (*p* < 0.05) compared to raw SLBH; T: temperature.

**Table 11 molecules-27-07243-t011:** Total phenolic content of raw and dehydrated stingless bee honey (SLBH).

Method of Dehydration		SLBH Species	T (°C)/ PL	Time	Total Phenolic Content (mg GAE/100 g)		Alteration	Author
					Raw	Dehydrated		
Thermal treatment		*Melipona bicolor*	90 °C	15–60 s	20.26	25–30 *	↑	[16]
		*M. bicolor*	95 °C	15–60 s	20.26	20–30 *	↑	
		*-*	50–90 °C	-	5130	5700–6750 *	↑	[23]
		*-*	45–90 °C	30–120 min	44.36	47.35–68.43	↑	[22]
Thermosonication		*-*	45–90 °C	30–120 min	47.50	49.02–75.08	↑	
Vacuum	Drying (5% moisture)	*Heterotrigona itama*	40 °C	-	21.31	20–25	↑	[21]
		*H. itama*	50–60 °C	-	21.31	20–35 *	↑	
	Drying (11% moisture)	*H. itama*	40–50 °C	-	21.31	20–25	↑	
		*H. itama*	60 °C	-	21.31	25–30 *	↑	
	Evaporation (11% moisture)	*H. itama*	40 °C	-	21.31	15–20 *	↓	
		*H. itama*	50 °C	-	21.31	20–25	↑	
		*H. itama*	60 °C	-	21.31	25–30 *	↑	
Freeze-drying (5% moisture)		*H. itama*	−54 °C	24 h	21.31	20–25	↑	
Microwave heating		*H. itama*	20 PL	15–30 s	12.45	11.93–12.11	↓	[18]
		*H. itama*	20 PL	60 s	12.45	13.45	↑	
		*H. itama*	60 PL	25–30 s	12.45	13.87–13.94	↑	
		*H. itama*	60 PL	60 s	12.45	17.9 *	↑	
		*H. itama*	100 PL	5–10 s	12.45	11.05–13.43	↓	
		*H. itama*	100 PL	15 s	12.45	14.02	↑	
Dehumidification		*H. itama*	35 °C	1 day	12.45	11.73 *	↓	
		*H. itama*	35 °C	2 days	12.45	12.70 *	↑	
Food dehydrator		*H. itama*	40 °C	12–84 h	41.99	57.83	↑	[17]
		*H. itama*	55 °C	12–84 h	41.99	73.77	↑	
		*H. itama*	70 °C	12–84 h	41.99	157.32	↑	
MARDI dehydrator		*H. itama*	30 °C	8 h	24.47	25	↑	[24]

* a significant difference (*p* < 0.05) compared to raw SLBH; mg GAE/100 g: milligram of gallic acid equivalents per 100 g of honey; T: temperature; PL: power level; ↑: increase; ↓: decrease.

**Table 12 molecules-27-07243-t012:** Total flavonoid content of raw and dehydrated stingless bee honey (SLBH).

Method of Dehydration		SLBH Species	T (°C)	Time	Total Flavonoid Content (mg QE/g)		Alteration	Author
					Raw	Dehydrated		
Thermal treatment		-	50	-	32.2	33.70 *	↑	[23]
		-	75	-	32.2	35.30 *	↑	
		-	90	-	32.2	36.43 *	↑	
Vacuum	Drying (5% moisture)	*Heterotrigona itama*	40–60	-	0.2–0.25	0.25–0.3 *	↑	[21]
	Drying (11% moisture)	*H. itama*	40	-	0.2–0.25	0.2–0.25	↑	
		*H. itama*	50–60	-	0.2–0.25	0.2–0.25 *	↑	
	Evaporation (11% moisture)	*H. itama*	40–60	-	0.2–0.25	0.2–0.25 *	↑	
Freeze-drying (5% moisture)		*H. itama*	−54	-	0.2–0.25	0.2–0.25 *	↑	

* a significant difference (*p* < 0.05) compared to raw SLBH; mg QE/g: milligram of quercetin equivalent per gram of hone; T: temperature; ↑: increase.

**Table 13 molecules-27-07243-t013:** Individual phenolic compounds of raw and dehydrated stingless bee honey (SLBH).

Method of Dehydration	SLBH Species	T (°C)	Time	Chlorogenic Acid (μg/100 g)		Alteration	Rosmarinic Acid (μg/100 g)		Alteration	Author
				Raw	Dehydrated		Raw	Dehydrated		
Thermal treatment	*Melipona bicolor*	90	15 s	<LOQ	11.7	↑	<LOQ	<LOQ	-	[16]
			60 s	<LOQ	9.59	↑	<LOQ	<LOQ	-	
	*M. bicolor*	95	15 s	<LOQ	12.3	↑	<LOQ	<LOQ	-	
			60 s	<LOQ	10.1	↑	<LOQ	<LOQ	-	
Vacuum drying	*Heterotrigona itama*	40	-	143.51	122.49	↓	600.86	623.36	↑	[21]
	*H. itama*	50	-	143.51	150.45	↑	600.86	725.64	↑	
	*H. itama*	60	-	143.51	153.47	↑	600.86	804.79	↑	
Freeze-drying	*H. itama*	−54	24 h	143.51	166.28	↑	600.86	768.98	↑	

LOQ: limit of quantification; T: temperature; ↑: increase; ↓: decrease.

**Table 14 molecules-27-07243-t014:** Individual phenolic compounds of raw and dehydrated stingless bee honey (SLBH).

Method of Dehydration	SLBH Species	T (°C)	Time	Rutin (μg/100 g)		Alteration	Quercetin (μg/100 g)		Alteration	Author
				Raw	Dehydrated		Raw	Dehydrated		
Thermal treatment	*Tetragonisca angustula*	52	470 min	ND	ND	-	41.28	9.63	↓	[15]
	*T. angustula*	55	170 min	ND	ND	-	41.28	11.47	↓	
	*T. angustula*	57	60 min	ND	ND	-	41.28	8.50	↓	
	*T. angustula*	60	22 min	ND	56.98	↑	41.28	81.18	↑	
	*T. angustula*	66	8 min	ND	43.08	↑	41.28	60.27	↑	
	*T. angustula*	66	3 min	ND	38.35	↑	41.28	48.10	↑	
	*T. angustula*	68	1 min	ND	27.51	↑	41.28	50.15	↑	
	*T. angustula*	71	24 s	ND	39.44	↑	41.28	51.15	↑	
Vacuum drying	*Heterotrigona itama*	40	-	82.88	67.25	↓	498.25	504.95	↑	[21]
	*H. itama*	50	-	82.88	73.82	↓	498.25	584.71	↑	
	*H. itama*	60	-	82.88	79.78	↓	498.25	646.72	↑	
Freeze-drying	*H. itama*	−54	24 h	82.88	90.82	↑	498.25	618.39	↑	

ND: not detected; T: temperature; ↑: increase; ↓: decrease.

**Table 15 molecules-27-07243-t015:** The optimal setting for each dehydration methods and impact on physicochemical properties of Stingless bee honey (SLBH).

Method of Dehydration		T (°C)/ PL	Time	MC (%)	WR (%)	WA	HMF (mg/kg)	pH	FA	Ash	EC	DA	TSS	TRC	TPC	TFC
Thermal treatment		90–95 °C	15–60 s	>17	4.2	-	<LOQ	NC	NC	-	NC	NC	↑	↓	↑	↑
		45–90 °C	30–120 min	>17	6.9	>0.6	42.40	-	-	-	-	-	-	-	↑	-
Thermosonication		45–90 °C	30–120 min	>17	16.6	>0.6	62.46	-	-	-	-	-	-	-	↑	-
Vacuum drying (5% moisture)		60 °C	-	<17	84.3	<0.6	12.18	↑	↓	-	-	-	-	↑	↑	↑
Freeze-drying (5% moisture)		−54 °C	24 h	<17	84.3	<0.6	9.29	NC	NC	-	-	-	-	↓	↑	↑
Microwave heating		60 PL	60 s	<17	52	-	-	NC	-	-	-	-	↑	-	↑	-
Dehumidification		35 °C	2 days	>17	45	-	-	NC	-	-	-	-	↑	-	↑	-
Food dehydrator		55 °C	18 h	<17	80	<0.6	<5.81	-	-	-	-	NC	-	-	↑	-
MARDI dehydrator		30 °C	8 h	>17	35	-	2.39	-	-	-	-	-	-	↑	↑	-
Clay pot storage	Large surface area	25 °C	10 days	>17	10.9	>0.6	-	NC	NC	↑	-	-	↑	↑	-	-
		35 °C	3 days	>17	24.2	>0.6	-	↑	NC	NC	↑	-	↑	-	-	-

NC: no changes of value in the parameter; ↑: increase; ↓: decrease; T: temperature; PL: power level; MC: moisture content.; WR: water reduction; WA: water activity; HMF: hydroxymethylfurfural; FA: free acidity; EC: electrical conductivity; DA: diastase activity; TSS: total soluble solids); TRC: total reducing sugar; TPC: total phenolic content; TFC: total flavonoid content.

## Data Availability

Not applicable.

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
