# Peer review of "Methods of the Dehydration Process and Its Effect on the Physicochemical Properties of Stingless Bee Honey: A Review"

_molecules, 2022, doi:10.3390/molecules27217243_

Round 1

Reviewer 1 Report

The topic on which the article focuses is very interesting and outlines the effects of treatments carried out on a type of honey on the quality and preservation of the product's original characteristics.

The article is well organized although I suggest revising the English.

Minor comments:

Page 2 line 17 “Apis” should be written in italics.

All tables are not mentioned in the text. Please quote them.

In the tables at least once I would write the name of the SLBH species not abbreviated so that those who are not 'insiders' understand what we are talking about.

I do not agree with putting bibliographical references concerning an entire paragraph. I generally believe that each sentence should have its own bibliographical reference, but in this work, it could be corrected.

Page 10 line 6 (below the table) “Apis” should be written in italics.

Page 18 table number should be 10 instead of 1

Reviewer 2 Report

The manuscript presents methods of the dehydration process and its effect on the physiochemical properties of stingless bee honey. However, given the number of types of SLBH species presented, the manuscript refers to a specific harvesting area, the area that is not specified. Is this the reason why only a few bibliographic references are referred to in the tables or is the manuscript focused only on these types? The introduction can be improved with general data on the physico-chemical and biological properties of this type of honey, a comparison with honey obtained from the genus Apis mellifera in order to emphasize the importance of these processing methods.The manuscript refers to the physico-chemical properties that can be modified during the dehydration activity. No data are presented regarding the composition of biologically active substances such as polyphenols (representative compounds), volatile substances that may undergo transformations during these processes. These data, if any, should be presented. In my opinion, the manuscript should be restructured, it is difficult to follow the data presented, many of the values presented in the tables are repeated in the text.

Reviewer 3 Report

Comments and Suggestions for Authors:

The present work is well written, but in my opinion, you should refer to more scientific papers, because it is an review article.

In paragraph 2.1. Moisture content some references are missing and they are not visible… Error! Reference source not found… Check out through the whole paper

A study by Yegge et al. has shown that the dehydration process using microwave…please insert the reference number after Yegge et al. [xxx]. Check out through the whole paper

Please technically improve Table 1.

Please technically improve Table 13.

Round 2

Reviewer 2 Report

In my opinion, the authors have reorganized the manuscript in a form suitable for the proposed purpose. I think that some minor corrections are necessary to give the desired meaning of some phrases. E.g:

Based on Table 1, several studies have shown

I think it would be more appropriate:

Several studies summarized/presented in table 1 showed that...

or

Based on Table 1, tThe dehydration process using thermal treatment....

I think it would be more appropriate:

As a conclusion, according to the data presented in table 1...

These corrections should be made throughout the manuscript where necessary.
